# Associations between Sleep Quality and Heart Rate Variability; Implications for a Biological Model of Stress Detection Using Wearable Technology

**DOI:** 10.3390/ijerph19095770

**Published:** 2022-05-09

**Authors:** Taryn Chalmers, Blake A. Hickey, Philip Newton, Chin-Teng Lin, David Sibbritt, Craig S. McLachlan, Roderick Clifton-Bligh, John W. Morley, Sara Lal

**Affiliations:** 1Neuroscience Research Unit, School of Life Sciences, University of Technology Sydney, Broadway, Sydney, NSW 2007, Australia; blake.hickey1@my.nd.edu.au (B.A.H.); sara.lal@uts.edu.au (S.L.); 2School of Nursing and Midwifery, Western Sydney University, Penrith, NSW 2747, Australia; p.newton@westernsydney.edu.au; 3Australian AI Institute, University of Technology Sydney, Broadway, Sydney, NSW 2007, Australia; chin-teng.lin@uts.edu.au; 4School of Public Health, University of Technology Sydney, Broadway, Sydney, NSW 2007, Australia; david.sibbritt@uts.edu.au; 5Centre for Healthy Futures, Torrens University, Sydney, NSW 2009, Australia; craig.mclachlan@laureate.edu.au; 6Kolling Institute for Medical Research, Royal North Shore Hospital, St Leonards, NSW 2064, Australia; roderick.cliftonbligh@sydney.edu.au; 7School of Medicine, Western Sydney University, Penrith, NSW 2747, Australia; j.morley@westernsydney.edu.au

**Keywords:** rate variability, sleep, stress, wearable technology

## Abstract

Introduction: The autonomic nervous system plays a vital role in the modulation of many vital bodily functions, one of which is sleep and wakefulness. Many studies have investigated the link between autonomic dysfunction and sleep cycles; however, few studies have investigated the links between short-term sleep health, as determined by the Pittsburgh Quality of Sleep Index (PSQI), such as subjective sleep quality, sleep latency, sleep duration, habitual sleep efficiency, sleep disturbances, use of sleeping medication, and daytime dysfunction, and autonomic functioning in healthy individuals. Aim: In this cross-sectional study, the aim was to investigate the links between short-term sleep quality and duration, and heart rate variability in 60 healthy individuals, in order to provide useful information about the effects of stress and sleep on heart rate variability (HRV) indices, which in turn could be integrated into biological models for wearable devices. Methods: Sleep parameters were collected from participants on commencement of the study, and HRV was derived using an electrocardiogram (ECG) during a resting and stress task (Trier Stress Test). Result: Low-frequency to high-frequency (LF:HF) ratio was significantly higher during the stress task than during the baseline resting phase, and very-low-frequency and high-frequency HRV were inversely related to impaired sleep during stress tasks. Conclusion: Given the ubiquitous nature of wearable technologies for monitoring health states, in particular HRV, it is important to consider the impacts of sleep states when using these technologies to interpret data. Very-low-frequency HRV during the stress task was found to be inversely related to three negative sleep indices: sleep quality, daytime dysfunction, and global sleep score.

## 1. Introduction

Autonomic control of inter-beat heart rate is modulated by the two branches of the autonomic nervous system: the sympathetic and parasympathetic arms. As such, analysis of fluctuations in heart rate over time, obtained via a non-invasive electrocardiogram, provides useful information regarding autonomic functioning. This measurement is known as heart rate variability (HRV). The clinical utility of heart rate variability was first realised in the late 1980s, when reduced HRV was found to be a robust clinical predictor of mortality post myocardial infarction [1]. Since then, many disease states have been investigated using heart rate variability. In particular, diseases that are associated with impaired or dysregulated autonomic control, such as renal failure, diabetes, and heart failure, have been linked with reduced or impaired HRV.

The autonomic nervous system plays a vital role in the modulation of many vital bodily functions, one of which is sleep and wakefulness. Many studies have investigated the link between autonomic dysfunction and sleep cycles; however, few studies have investigated the links between short-term sleep health, as determined by sleep length, quality, etc., and autonomic functioning in healthy individuals. Oftentimes, research has focused on cohorts suffering from insomnia and/or autonomic impairment [2,3,4], or used endpoints known to be predictors of impaired autonomic control, such as hypertension [5,6,7,8] or obstructive sleep apnoea [9,10,11,12,13], rather than direct assessment of short-term sleep quality and HRV. The literature has suggested that impaired autonomic functioning is associated with chronic insomnia, which may reflect physiological hyperarousal. This is supported by the current theoretic models for sleep initiation and maintenance disorders [14]. Research, however, is unclear as to whether isolated sleep disturbances influence autonomic function. Of the studies that have been conducted [15,16,17,18,19,20], there remains no clear consensus regarding the link between sleep quality and HRV parameters, in particular during resting and stress states.

The human body responds to acute stress via a cascade of hormonal [21,22], immune-mediated [23,24,25], and autonomic [26,27,28] processes that culminate in an acutely advantageous biological response. This transient response is generally short-lived and provides a momentary reservoir of biological resources that can be utilised in order to mitigate the internal or external stressor [28]. Over time, repeated exposure to stressful events can induce chronic activation of the stress response, via the hypothalamic–pituitary–adrenal (HPA) axis [29]. This chronic hypothalamic–pituitary–adrenal-axis (HPA-axis) activation has been implicated in numerous disease processes, such as depression [30], heart disease [31], hypertension [32], and cancer [33,34,35,36]. Although research has generally focused on the links between impaired sleep and cardiac health in patients with various diseases, little research has been conducted into the acute impacts of impaired sleep quality on cardiac parameters within a healthy population. This is particularly pertinent research when we consider the current widespread availability of technologies that can record both cardiac and sleep data. Numerous wearable technologies exist with the capabilities of recording HRV data; however, the real-time clinical significance of this information is yet to be fully elucidated. Further, the impact of stress on HRV, and the effect of sleep quality on HRV indices, remain important considerations when utilising HRV as a real-time predictor of stress and health.

In the current study, the aim was to investigate the links between sleep quality and duration, and HRV in 60 healthy participants, in order to provide useful information about the effects of stress and sleep on HRV indices. The a priori hypotheses were that (1) global sleep score would be inversely related to high-frequency HRV during both resting and stress tasks; (2) daytime dysfunction would be inversely related to resting very-low-frequency heart rate variability; (3) reported sleep quality during the stress task would be inversely related to very-low-frequency (VLF) HRV.

## 2. Materials and Methods

### 2.1. Participants and Sampling

A cross-sectional study design was used to examine the relationship between HRV and sleep quality in 60 healthy participants aged between 18–45 years. Participants were recruited from the general community via online advertisements. Prior to inclusion, an in-house-designed lifestyle questionnaire adapted from the Lifestyle Appraisal Questionnaire (LAQ) [37] was utilised to screen participants for medication use, alcohol intake (over 1.6 standard drinks a day), smoking habits (over 10 cigarettes a day), and chronic disease/illness. Participants were excluded if they answered affirmatively to any of the screening questions mentioned above. Participants were asked to refrain from consuming coffee and alcohol for 24 h prior, and from smoking cigarettes for 12 h prior to the study. Further, participants were required to have had a minimum of 8 h sleep in the evening preceding the study.

Two participants were excluded from the study, one due to taking a calcium channel blocker, and one due to the inability to stay for the duration of the study, with two further participants recruited to fill the gaps; there were no further participants excluded. The total number of participants was 62, with analysis conducted on 60 participants (Figure 1). The study had institutional ethics approval from the University of Technology Sydney Human Research Ethics Committee, and all participants provided signed informed consent prior to the commencement of the study.

### 2.2. Experimental Procedures

At the commencement and conclusion of each session, blood pressure (BP) measurements were taken three times with an automated blood pressure monitor (Omron IA1B, Kyoto, Japan), and averaged to confirm inclusion into the study (BP < 160/100 mmHg). Following BP collection, in-house-designed questionnaires were used to collect demographic, lifestyle, and work-related data. Additionally, the general health questionnaire (GHQ) was also utilised to assess general psychological wellbeing whilst the Pittsburgh Quality of Sleep Index was employed to assess self-reported sleep quality. Nineteen individual items are compiled to produce seven “component” scores: subjective sleep quality, sleep latency, sleep duration, habitual sleep efficiency, sleep disturbances, use of sleeping medication, and daytime dysfunction. The addition of these scores for each of the seven components yields a global PSQI score. The PSQI has been shown to yield strong internal homogeneity, consistency (test–retest reliability), and validity [38].

The Trier Social Stress Test (TSST) [39] was then utilised to elicit a controlled stress response in participants. For this section of the study, participants underwent a 15 min resting baseline session, followed by the TSST, which consisted of a 5 min preparation/anticipation task where participants were required to prepare a short speech, followed by a 5 min public speaking task, and finally a 5 min mental arithmetic task. The TSST is one of the most well-accepted laboratory techniques to induce acute stress in experimental settings and has been repeatedly shown to reliably provoke hypothalamic–pituitary–adrenal-axis activation [39,40,41,42,43,44,45]. In fact, of the numerous stress-inducing laboratory protocols, meta-analysis has suggested that TSST is the most useful and appropriately standardised protocol for studies of stress assessment and stress hormone reactivity [40].

During baseline and each component of the TSST, 3-lead ECG data were captured using disposable Ag/AgCl electrodes placed on the participant’s upper torso under each clavicle on the coracoid processes, and one just below the sternum over the xiphoid process [46]. Following completion of the TSST, three further BP readings were obtained, which were then averaged.

### 2.3. ECG Data Processing

Raw ECG data were processed utilising the Kubios HRV Premium software (Version 3.1.0, Kubios Oy, Kuopio, Finland) to generate HRV parameters. Frequency domain HRV deconstructs the variance in beat-to-beat heart rate (HR) into its underlying components at different frequencies using fast-Fourier transforms (FFTs). It should be highlighted that the calculation of frequency domain HRV requires a certain level of “stationarity”, i.e., the mean and variance of the ECG signal do not vary significantly at different points during the electrocardiogram. In order to meet this requirement, data were collected over 5 min intervals and averaged to produce reliable HRV frequency domain data. Further, all data were assessed to ensure sinus rhythm prior to transformation. Frequency domain activity was calculated using Welch’s periodogram method [47] for the following HRV frequency bands: low-frequency (LF) power (0.04–0.15 Hz), high-frequency (HF) power (0.15–0.4 Hz), and total power (TP), as well as the LF:HF ratio. It should be noted that in the process of deriving these variables, the automatic artefact correction process of the Kubios software [48], as well as the smoothness priors method [49] of trend component rejection, were utilised, and HRV data were log-transformed prior to analysis where relevant.

### 2.4. Statistical Analysis

Statistical analysis was performed using SPSS Version 23.0 (SPSS Inc.; Chicago, IL, USA) with statistical significance defined as *p* < 0.05. Data were initially subject to descriptive statistics. Dependent sample *t*-tests were utilised in order to determine significant differences in HRV parameters between the resting phase and stress task. As multiple comparisons were made, a Holm–Bonferroni correction was then applied to avoid type I errors with the family-wise α (a) level set at 0.05. Following that, partial Pearson’s correlation analysis (controlling for age and body mass index (BMI)) was utilised to determine associations between HRV parameters and sleep factors (as determined by the PSQI) during the baseline and stress task (TSST). Where three or more significant relationships to a dependent variable were found (after Bonferroni correction), a multiple linear regression analysis was utilised to further examine the relationship between HRV and dependent variables.

## 3. Results

### 3.1. Participant Descriptions

This study consisted of 60 participants (male *n* = 33; female *n* = 27), with demographic information provided in Table 1. The average age of participants was 28.9 ± 8.8 (years ± SD), with an average body mass index of 23.1 ± 3.4 kg/m^2^. Average pre-study (baseline systolic blood pressure (SBP) was 116.0 ± 14.4, and pre-study diastolic blood pressure (DBP) was 76.1 ± 9.2. Average post study SBP was 119.5 ± 13.6 and post-study DBP was 80.7 ± 8.81.

### 3.2. Comparison in Cardiac Parameters between Baseline and Stress Task

Significant differences were identified between certain cardiac parameters during the baseline (resting) phase and the stress task (Table 2). SBP (*p* = 0.001), DBP (*p* ≤ 0.001), LF (*p* = 0.006), HF (*p* ≤ 0.001), and ratio (*p* = 0.012) were significantly higher during the stress task.

### 3.3. Correlations between Sleep Quality and HRV

Partial correlations (controlling for age and BMI) were undertaken between HRV parameters during the baseline (resting) phase and stress task and self-reported sleep qualities (as measured by the PSQI) (Table 3). No significant correlations were identified during the baseline (resting) phase and the various subscales of the PSQI. During the stress task, VLF HRV was inversely correlated to daytime dysfunction (*p* = 0.001), subjective sleep quality (*p* = 0.007), and global sleep score (*p* = 0.018). HF n.u. during the stress task was inversely correlated to sleep latency (*p* = 0.013) and global sleep score (*p* = 0.014). Total power during the stress task was inversely correlated to daytime dysfunction (*p* = 0.003), subjective sleep quality (*p* = 0.014), and global sleep score (*p* = 0.019). Finally, LF:HF ratio during the stress task was positively correlated to sleep latency (*p* = 0.045).

### 3.4. Multiple Regressions

A multiple regression was performed when three or more independent variables were correlated to one dependent variable (after a Bonferroni correction). Daytime dysfunction, subjective sleep quality, and global sleep score significantly predicted VLF during the stress task (F(3, 55) = 2.84, *p* = 0.046, R = 0.37, R^2^ = 0.143; adjusted R^2^ = 0.09). Together, the independent variables explain 36.6% of the variability in VLF during the stress task (Table 4). None of the individual variables added significantly to the prediction.

After Bonferroni correction, daytime dysfunction, subjective sleep quality, and global sleep scores were entered into a multiple linear regression to predict TP during the stress task; however, the model was not significant (F(3, 55) = 2.5, *p* = 0.073, R = 0.34, R^2^ = 0.12; adjusted R^2^ = 0.07).

## 4. Discussion

The present study examined relationships between sleep quality indices as determined by the PSQI (subjective sleep quality, sleep latency, sleep duration, habitual sleep efficiency, sleep disturbances, use of sleeping medication, and daytime dysfunction) and heart rate variability parameters during resting and stress states. Very-low-frequency HRV during the stress task was found to be inversely related to three negative sleep indices: sleep quality, daytime dysfunction, and global sleep score. That is, the poorer an individual’s sleep quality, and the greater the effect on their daytime function, the lower the VLF HRV during the stress task. Many studies have suggested that the renin–angiotensin system contributes significantly to VLF [50,51,52]. Further, studies have shown that impaired sleep (i.e., obstructive sleep apnoea) is linked with nocturnal renin–angiotensin-system (RAS) activation [53,54], and thus may be reflected in increased nocturnal VLF HRV [55]. Nocturnal RAS activation has subsequently been shown to reduce daytime RAS activation, which may account for the findings of this study [56]. Individuals who experience nocturnal RAS activation may then exhibit augmented RAS activation in response to stress, resulting in lower VLF HRV.

This study also identified significant rise in systolic and diastolic blood pressure during the stress task, when compared with the resting state. It is well accepted that the TSST reliably increases cardiovascular activity, such as heart rate and blood pressure.

LF:HF ratio was significantly higher during the stress task than during the baseline resting phase. Research supports an increase in sympathetic nervous system activation during a stress task, and a synchronised parasympathetic withdrawal [57]. The sympathetic shift proves advantageous during the stress response, mobilising biological reserves, increasing blood flow to large muscle groups and vital organs, and promoting efficient respiratory and cardiac function [58].

High-frequency HRV was found to be inversely related to sleep latency and global sleep score during the stress task. This may suggest that individuals who experience impaired night-time sleep exhibit a sympathetic predominance during waking hours. This is supported by the literature, with studies showing that compromised sleep durations and patterns are associated with higher daytime heart rate [59], blood pressure [60], and stress [61].

Due to the cross-sectional nature of the study, only short-term variations in HRV could be assessed. Further research implementing longer-term measurements in HRV could be a value. This study utilised the PSQI to assess sleep disturbance in the previous one month. Future research may benefit from assessment of an individual’s sleep habits over a longer period of time, and, further, by utilising objective sleep measures such as polysomnography. A larger sample size may also increase the robustness of future studies.

A limitation of the study was that ECG data for HRV were collected over 5 min intervals and averaged for calculation of frequency domain HRV to establish stationarity. Given that data are averaged, significant data reflective of short-term stress may have not been accounted for, which, in the context of sleep, is important due to short blips of stress-related HRV changes during sleep, such as with night-terrors.

## 5. Conclusions

The aim of this study was to investigate the possible impacts of short-term sleep disturbances on HRV parameters during resting and stress states. The present study identified unique relationships between impaired sleep and stress-state HRV. Given the ubiquitous nature of wearable technologies for monitoring health states, it is important to consider the impacts of varied biology conditions when using these technologies to interpret data.

## Figures and Tables

**Figure 1 ijerph-19-05770-f001:**
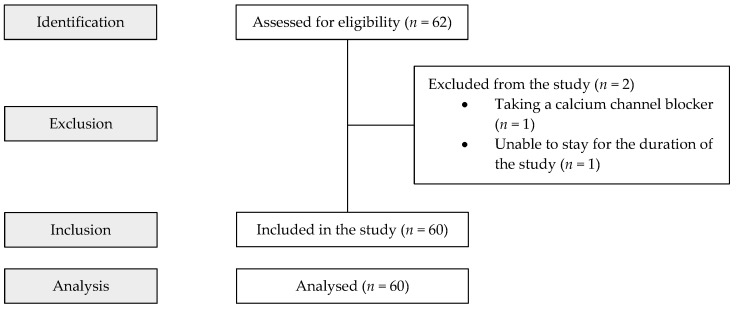
STROBE participant flow diagram.

**Table 1 ijerph-19-05770-t001:** Demographics of study participants (*n* = 60).

Measure	Participants (*n* = 60) Value ± SD (Range)
Male gender (%)	55.0 (Male 33; Female 27)
Age (years)	28.9 ± 8.8 (19–70)
Height (cm)	174.5 ± 9.8 (154–192)
Weight (kg)	72.7 ± 14.8 (52–138)
BMI (kg/m^2^)	23.1 ± 3.4 (18–35)
Pre-study SBP	116.0 ± 14.4 (78–143)
Pre-study DBP	76.1 ± 9.2 (56–92)
Post-study SBP	119.5 ± 13.6 (76–149)
Post-study DBP	80.7 ± 8.81 (61–102)

Key: BMI = body mass index; DBP = diastolic blood pressure; SBP = systolic blood pressure; SD = standard deviation.

**Table 2 ijerph-19-05770-t002:** Comparison of cardiac parameters between baseline (resting) and stress task (*n* = 60).

Parameter	Resting Phase	Stress Task	*p*
SBP	116.0 ± 14.4	119.5 ± 13.6	0.001 *
DBP	76.1 ± 9.2	80.7 ± 8.8	<0.001 *
VLF	4.6 ± 0.7	4.4 ± 0.9	0.143
LF n.u.	53.8 ± 3.2	54.8 ± 2.7	0.006 *
HF n.u.	46.2 ± 3.2	50.8 ± 6.6	<0.001 *
TP	17.2 ± 2.5	17.0 ± 2.7	0.421
Ratio	1.2 ± 0.2	1.3 ± 0.1	0.012 *

* Statistical significance: *p* < 0.05. Key: DBP = diastolic blood pressure; HF n.u. = high-frequency heart rate variability (normalised units); LF n.u. = low-frequency heart rate variability; *p* = significance; ratio = sympathovagal balance; SBP = systolic blood pressure; TP = total power heart rate variability; VLF = very-low frequency heart rate variability.

**Table 3 ijerph-19-05770-t003:** Partial correlations (controlling for age and BMI) between sleep quality (as measured by the PSQI) and heart rate variability (*n* = 60).

Control Variables: Age and BMI		Sleep Latency	Daytime Dysfunction	Subjective Sleep Quality	Global Sleep Score
Stress VLF	Correlation	−0.143	−0.448	−0.361	−0.320
	Significance (2-tailed)	0.303	0.001 *	0.007 *	0.018 *
	df	52	52	52	52
Stress HF n.u.	Correlation	−0.334	−0.166	−0.255	−0.332
	Significance (2-tailed)	0.013 *	0.231	0.062	0.014 *
	df	52	52	52	52
Stress TP	Correlation	−0.184	−0.399	−0.332	−0.318
	Significance (2-tailed)	0.183	0.003 *	0.014 *	0.019 *
	df	52	52	52	52
Stress Ratio	Correlation	0.274	0.094	0.094	0.158
	Significance (2-tailed)	0.045 *	0.499	0.498	0.253
	df	52	52	52	52

* Statistical significance: *p* < 0.05. Key: BMI = body mass index; df = degrees of freedom; HF n.u. = high-frequency heart rate variability (normalised units); ratio = sympathovagal balance; TP = total power heart rate variability; VLF = very-low-frequency heart rate variability.

**Table 4 ijerph-19-05770-t004:** Multiple regression analysis VLF during the stress task.

Regression Analysis VLF (Stress)—R = 0.37; R^2^ = 0.134; Adjusted R^2^ = 0.087; F = 2.84; *p* = 0.046 *					
Variable	B	SE	β	t	*p*
Daytime dysfunction	−0.269	0.19	−0.20	−1.44	0.15
Subjective sleep quality	−0.340	0.24	−0.23	−1.41	0.17
Global sleep score	−0.008	0.06	−0.02	−0.123	0.90

* Statistical significance: *p* < 0.05. Key: B = unstandardised regression coefficient; β = standardised coefficient; *p* = level of statistical significance; R = multiple correlation coefficient; R^2^ = proportion of variance; SE = standard error; t = t statistic; VLF = very-low-frequency heart rate variability.

## Data Availability

The data presented in this study are available on request from the corresponding author. The data are not publicly available currently due to storage on site at the University of Technology, as stipulated in the Ethics application.

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
