# Peer review of "Associations between Sleep Quality and Heart Rate Variability: Implications for a Biological Model of Stress Detection Using Wearable Technology"

_ijerph, 2022, doi:10.3390/ijerph19095770_

Round 1

Reviewer 1 Report

The authors focus on the statistical verification of the hypotheses they defined in the introduction. But on what basis did they establish hypothesis (2)? Is there any premise that implies causality in this matter?

In chapter 2.3. states that data was collected over 5-minute intervals and averaged. Can't significant data appear just like short-term sleep disorders due to stress? (For example, nightmares). Would averaging over a 5-minute interval result in the loss of critical information for these excesses? I think these issues could at least be mentioned in the discussion.

In the abstract - line 30 "collected 29 form participants" - should it be "from"?

Author Response

The authors focus on the statistical verification of the hypotheses they defined in the introduction. But on what basis did they establish hypothesis (2)? Is there any premise that implies causality in this matter?

Response: Hypothesis (2) daytime dysfunction would be inversely related to resting very low frequency heart rate variability. VLF amplitude is dampened by upregulated SNS, as would be seen in stress (Shaffer, F., & Ginsberg, J. P. (2017). An overview of heart rate variability metrics and norms. Frontiers in public health, 258.); therefore, it was postulated that stress leading to daytime dysfunction through poor sleep, will be reflected in low VLF readings. The VLF component has been previously considered to be a marker of hormonal fluctuations (e.g. testosterone) though its relationship to other hormones is unclear/not established Tobaldini, E., Nobili, L., Strada, S., Casali, K. R., Braghiroli, A., & Montano, N. (2013). Heart rate variability in normal and pathological sleep. Frontiers in physiology, 4, 294.). Interestingly, ultra low frequency is thought to be primarily driven by circadian rhythm and would be worth incorporating into future studies .

In chapter 2.3. states that data was collected over 5-minute intervals and averaged. Can't significant data appear just like short-term sleep disorders due to stress? (For example, nightmares). Would averaging over a 5-minute interval result in the loss of critical information for these excesses? I think these issues could at least be mentioned in the discussion.

Response: This is a good point and will be noted in limitations with some important data which may have been missed. However, As noted In methods, calculation of frequency domain HRV requires a certain level of “stationarity” and in order to meet this requirement, data was collected over 5-minute intervals and averaged to produce reliable HRV frequency domain data.

In the abstract - line 30 "collected 29 form participants" - should it be "from"?

Response: Yes, this has been corrected to ‘from’.

Reviewer 2 Report

The authors propose an investigation of the possible connections between sleep quality and heart rate variability. The analysis is interesting and well conducted, and the results are properly presented. The paper can be published in the present form, after just a minor revision:

  • Heart rate variability also depends on the specific sleep phase. For example, in REM sleep phase, HR is less variable. This is one of the principles used by wearables like FitBit to estimate REM sleep. Does this alterate, in some way, the HRV parameters used for the correlations?

Author Response

Response: This study unfortunately did not consider the specific phases of sleep when using HRV parameters in correlations; however, the authors are currently performing a new study which will assess phases of sleep. This is a limitation of the correlations performed.

Reviewer 3 Report

The authors approach an important but well-explored relationship between sleep and HRV as a potential reporter for stress related sleep issues. The intent to improve information extraction from easy-to-assess metrics like HRV to improve sleep health is lauded. Unfortunately the manuscript raises a number of methodological concerns, and does not provide a focused-enough description of its own novelty. More comparison to the (large) literature on sleep, stress, and HRV is needed; better descriptions of recruitment and comparison across individuals is needed; improved reporting and presentation of results is needed for useful interpretation.

I did not attempt an exhaustive edit, but the following comments are hopefully useful in guiding revision efforts:

What’s novel here? There are many papers on the topic of sleep, stress, and HRV. The authors need to be more clear in the abstract and introduction about what is actually novel here.

Intro. You state: “there remains no clear consensus regarding the link between sleep quality and HRV parameters, in particular during resting and stress states.” Citing the large literature and or reviews here would help establish what you are differentiating yourselves from.

“Healthy” is defined as up to 16 alcoholic drinks per day?! Is that a reasonable criteria? That level of consumption will have a large impact on sleep. Tables do not seem to reflect the level of “healthy” consumption in this cohort, but that is important to ascertain in this context, if these were your inclusion criteria.

You end the methods section 2.2 with “The total number of participants was 60.” But it is unclear if this is the result of the exclusions mentioned above or the total number from which individuals were then excluded. An initial and final count would be more appropriate, if not a flow chart capturing loss from recruitment by criterion.

Average systolic pressure is 119. Clinical elevation starts at 120. Again, for a “healthy” cohort, this is worrying. A more detailed breakdown of the correlations by group of individuals with similar characteristics is needed (or better yet, a mixed model to account for the ranges of related variables represented by this cohort).

Average within individual change would be more appropriate for Table 2 than populations means. Given the range of the cohort, the mean carries little information, whereas a table (and/or figure) conveying median change (or change across all individuals, in the figure) would allow me to interpret how big and or reliable the changes described really are. Capturing heterogeneous effects is an important part of exploring natural populations, so this could be highlighted as a feature, rather than seen as a confound.

The first two paragraphs of the discussion are just about the fact that expected patterns were found. I assume this is a validation of the approach taken, but it is not novel. I suggest leading with the novel finding (which I believe is the correlations in discussion paragraph 3). This would go along with other suggested changes above to improve interpretability of what the authors are saying is the useful, novel discovery they are sharing.

Author Response

What’s novel here? There are many papers on the topic of sleep, stress, and HRV. The authors need to be more clear in the abstract and introduction about what is actually novel here.

Response:  This research is novel in that it uses short-term sleep quality and HRV parameters.  This has rarely been examined and has been highlighted in the abstract and introduction.

Intro. You state: “there remains no clear consensus regarding the link between sleep quality and HRV parameters, in particular during resting and stress states.” Citing the large literature and or reviews here would help establish what you are differentiating yourselves from

Response: This has been updated.

“Healthy” is defined as up to 16 alcoholic drinks per day?! Is that a reasonable criterion? That level of consumption will have a large impact on sleep. Tables do not seem to reflect the level of “healthy” consumption in this cohort, but that is important to ascertain in this context, if these were your inclusion criteria.

Response: This was a typo – 1.6 STD/day; which has been amended.

You end the methods section 2.2 with “The total number of participants was 60.” But it is unclear if this is the result of the exclusions mentioned above or the total number from which individuals were then excluded. An initial and final count would be more appropriate, if not a flow chart capturing loss from recruitment by criterion.

Response:  Two participants were excluded from the study, one due to taking a calcium channel blocker, and one due to the inability to stay for the duration of the study, with two further participants recruited to fill the gaps; there were no further participants excluded.

Average systolic pressure is 119. Clinical elevation starts at 120. Again, for a “healthy” cohort, this is worrying. A more detailed breakdown of the correlations by group of individuals with similar characteristics is needed (or better yet, a mixed model to account for the ranges of related variables represented by this cohort).

Response: It was considered doing more in-depth correlations with respect to SBP; however, majority of SBP readings were less <139mmHg, with these numbers not considered pathological, merely high normal. Further, pre vs post stress test only unveiled a minute rise in SBP. We were unsure if further correlations by group would yield anything more significant.

The first two paragraphs of the discussion are just about the fact that expected patterns were found. I assume this is a validation of the approach taken, but it is not novel. I suggest leading with the novel finding (which I believe is the correlations in discussion paragraph 3). This would go along with other suggested changes above to improve interpretability of what the authors are saying is the useful, novel discovery they are sharing.

Response: This has been updated.